# Engaging Community Health Workers (CHWs) in Africa: Lessons from the Canadian Red Cross supported programs

Dina Idriss-Wheeler[1,2][¤a]*, Ilja Ormel[2], Mekdes Assefa[2‡], Faiza Rab[2‡], Christina Angelakis[2‡], Sanni Yaya[3,4‡], Salim Sohani[2][¤b]

1 Faculty of Health Sciences, Interdisciplinary School of Health Sciences, University of Ottawa, Ottawa, Ontario, Canada, 2 Health Intelligence Research and Development, Canadian Red Cross, Ottawa, Ontario, Canada, 3 Faculty of Social Sciences, School of International Development and Global Studies, University of Ottawa, Ottawa, Ontario, Canada, 4 The George Institute for Global Health, Imperial College London, London, United Kingdom

☯ These authors contributed equally to this work.
¤a Current address: Interdisciplinary School of Health Sciences, University of Ottawa, Ottawa, Ontario, Canada
¤b Current address: Canadian Red Cross, Ottawa, Ontario, Canada
‡ These authors also contributed equally to this work
* didri040@uottawa.ca

**Data Availability Statement:** Please note that we are unable to to provide the actual reports that were extracted for data publicly because of ethical

## Abstract

Universal Health Coverage (UHC) will not be achieved if health care worker shortages, estimated to increase to 18 million by 2030, are not addressed rapidly. Community-based health systems, which pivot to effective engagement of community health workers (CHW), may have an essential role in linking communities with health care facilities and reducing unmet health services needs caused by these shortages. The Canadian Red Cross (CRC) has partnered with different National Red Cross/Red Crescent Societies and Ministries of Health in Africa in the implementation of programs where CHWs contributed to the provision of various health services. This study reports on *key findings (i.e., beneficiaries reached, CHWs engaged, programs implemented, intervention outcomes) and lessons learned from CRC supported CHW programs in Africa over the last 15 years (2007–2022)*. Qualitative methodology was employed to conduct document analysis on 17 sets of reports from each CRC-supported community health worker project in Africa over the past 15 years. Focus was on identifying challenges, facilitators, and lessons learned. CRC supported projects have trained over 9000 CHWs, benefiting nearly 7.5 million people across Africa. Key success factors include adaptability and agility in programming and project management, and considering contextual factors (political, social, and cultural systems). Investing in essential training for CHWs, staff, and volunteers is crucial, alongside employing an evidence-based approach to inform all aspects of programming and implementation. Additionally, projects prioritizing protection, gender and inclusion (PGI) while leveraging existing community structures and partnerships important for successful implementation. Despite challenges (i.e., weak health systems, lack of political commitment, insufficient funding, inadequate training) CHWs are recognized as crucial in promoting community-based health, improving access to care, addressing disparities, and contributing to achieving (UHC). Their unique position

and legal restrictions on sharing sensitive data sets that are part of a secure database at the Canadian Red Cross. The database contains potentially identifying or sensitive participant information in the project base-line and end-line reports which are part of the reports used for extraction and analysis. We will make available the excel extraction tool used that outlines the name of the report and data extracted from the report for anyone interested. The figure also provides the analysis and qualitative coding of the data. For more information please contact the corresponding author and/or communications@redcross.ca to which additional data requests may be sent.

**Funding:** This work was supported in part by the Canadian Institutes of Health Research (CIHR)-Health System Impact (HSI) Doctoral Fellowship (202203IF1-483699-HID-CECA-167362) to DIW during her Doctoral Fellowship at the Canadian Red Cross from 2022-2023. The funders had no role in the study design, data collection and analysis, decision to publish, or preparation of the manuscript.

**Competing interests:** The authors have declared that no competing interests exist.

within communities enables them to provide culturally appropriate and localized primary health care- particularly in remote, resource limited and poverty-stricken regions.

## Introduction

Community Health Workers (CHWs) are an essential emerging global health workforce, bridging gaps in Universal Health Coverage (UHC) through access to the most difficult to reach and vulnerable populations [1]. Health workforce shortage remains a significant contributor to lack of access to healthcare [2]. A 2018 World Health Organization (WHO) report forecast a deficit of over 18 million health workers by 2030, with regions of Africa and Southeast Asia inequitably bearing the burden of these shortages [3,4]. Health workforce deficiencies in numbers, quality and performance continue to be a challenge in achieving UHC globally [5]. Climate change, migration, food insecurities, wars, infectious disease outbreaks and growing inequities are increasingly stressing already weakened systems, especially during humanitarian emergencies [6,7]. Without timely action, achieving UHC remains in peril. Therefore, viable, localized and sustainable solutions for hard-to-reach communities, while linking to formal health care systems in humanitarian settings are imperative.

The WHO defines CHWs as, "Health care providers who live in the community they serve and receive lower levels of formal education and training than professional health care workers such as nurses and doctors. This human resource group has enormous potential to extend health care services to vulnerable populations, such as communities living in remote areas and historically marginalized people, to meet unmet health needs in a culturally appropriate manner, improve access to services, address inequities in health status and improve health system performance and efficiency" [8] The roles and responsibilities of CHWs have evolved over time from community mobilization and health promotion to more complex preventive and some clinical services [2]. A 2020 WHO meta-review reported that CHWs perform a variety of tasks that can be organized into six main roles: (1) delivering diagnostic, treatment or clinical care, (2) encouraging uptake of health services, (3) providing health education and behaviour change motivation, (4) data collection and record-keeping, (5) improving relationships between health system functionaries and community members, and (6) providing psychosocial supports [8]. Additionally, CHWs have handled more complex work such as health counselling (i.e., HIV counselling) and providing basic medical treatment (i.e., integrated community care management (iCCM) of uncomplicated childhood diseases, delivering injectable contraceptives and conducting rapid malaria diagnosis) [8].

In humanitarian crises settings—in Honduras, Liberia and Kenya—CHWs were essential in ensuring a certain degree of the continuation of or access to basic health care services, mitigating the impact and spread of infectious diseases, empowerment of to make healthy maternal health choices and improving equity in terms of access to some services [9–12]. In some regions in Africa (listed below), CHWs were engaged in risk communication and community engagement, surveillance, contact tracing and referrals for testing [13]. Close to 80% of the population in the Sub-Saharan region, or 24 countries, rely heavily on CHW: Angola, Benin, Burkina Faso, Cote d'Ivoire, DRC, Ethiopia, Ghana, Kenya, Lesotho, Liberia, Niger, Nigeria, Madagascar, Mali, Malawi, Mozambique, Rwanda, Senegal, Sierra Leone, Uganda, Tanzania, Togo, Zambia, Zimbabwe [14]. The Africa Centre for Disease Control and Prevention (CDC) invested in CHWs through their Partnership to Accelerate COVID-19 Testing (PACT) is paving the way for funding across the region to increase the number of CHWs [4,13,15].

A 2020 scoping review of literature on health and nutritional services provided by CHWs in humanitarian settings revealed key findings, calling for incorporation of lessons learned from years of experience in the field [16]. The review highlighted CHW's ability to provide acute and prolonged services depending on context (i.e., protracted/acute) along with the need for community engagement and participation—including elders and community members in championing and providing the services–and emphasized the importance of flexible funding as well as access to necessary supplies (i.e., supply chain/buffer for commodities) and resources (i.e., mobile phones to input data/communicate during crisis and travel restrictions) [16]. In a recent systematic review by Werner et al. (2023), evidence suggested that interventions utilizing CHWs were effective as well as efficient in addressing barriers associated with accessing care in conflict-affected regions [17]. Similar to Miller et al. (2020), findings outlined how CHWs can use their closeness to the community and social ties to enhance healthcare by assisting people in their own communities to access care, detect diseases, and ensure they follow through with their treatment through iCCM [17]. Task shifting and task sharing or supervision with formally trained healthcare workers (i.e., health care staff in facilities) was also highlighted as a way to increase access to health services so more complex cases are referred/treated by health professionals [17].

These kinds of initiatives and literature demonstrate efforts to finding ways to fill the health workers gaps plaguing many health care systems in humanitarian, low-resource and conflict settings. While there is mounting evidence of effectiveness and versatility of CHWs with their ability to create culturally responsive interventions as well as community-based and local engagement [18], important knowledge gaps remain in how CHW programs can be best implemented in humanitarian, low resource and conflict settings.

The International Federation of the Red Cross and Red Crescent Societies (IFRC) is an international humanitarian network comprising over 192 National Red Cross and Red Crescent Societies (RCRC) [19]. The network has developed comprehensive guidelines outlining a strong case for community-based care and support programs to help strengthen health systems in LMICs as they work towards achieving UHC and reaching people who cannot use or access health services due to unaffordable or unavailable health services [19].

In 2022, the IFRC and African CDC signed a five-year program (2022–2026) called "Resilient and Empowered African Community Health" (REACH), with an anticipated budget of $2 billion, aimed at addressing the shortage of healthcare workers across Africa. The program aims to train and engage a total of two million CHWs over the course of six phases, beginning with Central African Republic (CAR), Egypt, Eswatini, Malawi, South Sudan, and the Republic of Congo. The average number of CHWs targeted for each country is approximately 36,363. In doing so, IFRC can leverage key strengths from its network of national societies who have (i) worked in numerous communities across Africa that are the most isolated and exposed to disasters and conflicts, (ii) partnered with government authorities, particularly ministries of health, in those regions to implement programs, and (iii) established a network of volunteers to reach the most isolated and vulnerable communities during crisis, and (iv) formed long-standing relationships through trust and rapport with local communities, increasing the likelihood of program uptake. The Canadian Red Cross (CRC) is one of these societies who has implemented community-based programming across Africa and other parts of the world for decades.

## Red cross and red crescent movement's contribution to CHW programing

Canadian Red Cross's (CRC) mission is "to help people and communities in Canada and around the world in times of need and support them in strengthening their resilience" [20].

More specifically, CRC's Health in Emergencies (HiE) unit prioritizes strengthening capacity of NS Societies and local stakeholders across the world to implement key projects and programs that strengthen health systems and resilience at the community level, while reinforcing the linkages with formal healthcare systems. Key initiatives are centered around community-based systems that include CHWs and other frontline lay health care workers. They play a unique role in hard to reach, fragile and conflicted affected regions of the world to help provide essential health care services. CRC best practices include partnering with local RCRC National Societies and ministries of health (or relevant government bodies), local structures to ensure contextual relevance and alignment with local formal health systems when implementing CHW related programs across several regions of Africa and Asia.

The organization has specifically carried out a range of projects in Africa, where CHWs worked in maternal, neonatal, and child health; water sanitation and hygiene (WASH); vaccination; Ebola Virus Disease recovery; emergency preparedness; and increased resilience to chronic drought conditions. These projects took place in Liberia, Kenya, Mali, Somalia, South Sudan, CAR, and Uganda. CRC has evaluated, and documented lessons learned from the implemented programs through monitoring data and reports which are housed in CRC's secure databases. The learning from these programs could make an important contribution to the successful implementation of CHW programs in humanitarian, low resource and conflict settings. This bridges the gap of health human resources in resource poor settings and improves access to essential care [21].

This study is relevant to the current contexts of humanitarian work in Africa. As the region ramps up to increase their CHW numbers and as IFRC and the African CDC implement the REACH program, this document review will inform program implementation by providing lessons learned by CRC and sharing decades of experience and expertise supporting CHW programs and working with National Societies. CRC's support and tenure in working on projects to support the implementation of CHW programs aligns well and can be leveraged to achieve the objectives of preparing CHWs in Africa to meet the health human resource gap. Additionally, findings from this study will contribute to addressing gaps in global knowledge and literature by providing actionable insights and evidence-based recommendations for the implementation of CHW programs in humanitarian and low-resource settings.

With this document review, we aim to extract key facilitators, barriers and lessons learned from CRC supported CHW programs in Africa over the last 15 years (2007–2022). The key objectives of the study are to:

• Identify major outcomes, challenges, solutions, facilitators, and lessons learned from CRC supported CHW programs.

• Outline relevant lessons learned for future CHW program implementation.

## Methods

A document analysis approach was used to extract and analyze the reports for each CRC supported CHW project implemented in Africa over the last 15 years (2007–2022). As described by Dalglish et al. (2020), this qualitative method uses the READ approach to document analysis [22]. Although this systematic procedure for collecting documents and gaining information is described in a health policy context at any level (i.e., global, national, local), the four steps were conducive and easily modifiable to analyze documents/reports produced from this study's projects implemented in humanitarian settings. The four steps are described below:

1. Ready your materials (document search and collection): the study includes documents in CRC's secure systems (PIMS and Data Repository) which house all CRC's proposals, interim reports, evaluation reports, ongoing monitoring data, baseline and endline survey data aggregate reports, and final reports for all projects implemented. The sample includes 17 project documents bounded by CRC supported CHW programs implemented and completed in Africa between 2007–2022 (15 years). The relevant documents were downloaded from CRC's Health in Emergencies secure data repository and housed in secure Microsoft Teams channel for data extraction by three members of the research team (MA, CA, DIW). This data was accessed between January 2023 –August 2023 for research purposes.

2. Extraction of Data: Team members extracted the data from the reports using a pre-determined extraction table as a guide to begin. The table had been created based on highlighting the key players in the project, key outcomes variable statistics, what the CHWs did in the program, if/how they were integrated into the formal health system, facilitators, barriers, lessons learned, and highlighted key protection, gender, equity, and inclusion aspects. The extractors began their extractions and re-adjusted the parameters accordingly. Each project was extracted by two members of the team and discrepancies/conflicts were resolved through discussion by the extractors. A third reviewer was consulted when agreement could not be reached.

3. Data Analysis: Once data were extracted, both qualitative and quantitative approaches were used to analyze the data. Descriptive statistics of key data parameters and outcomes were used to provide an overall picture of the impact of CHW projects. Thematic analysis, as described by Braun and Clarke (2006, 2019), was used to analyze the facilitators, barriers, lessons learned and recommendations. Two independent coders analyzed the reports for key features, collated codes into themes, met to discuss/review the themes and ensure they answer the key question of what lessons were learned (including facilitators, barriers) for implementing CRC supported CHW programs in Africa over the past 15 years.

4. Distil your Findings: Once analyzed, the data were presented using charts, graphs, tables, and diagrams.

Ethics approval was obtained from the University of Ottawa Office of Research Ethics and Integrity (H-01-13-8656) for this research project. This approval ensured that our work adhered to the highest ethical standards. Since no direct participants were included in this study, but rather existing administrative and research reports were analyzed, there was no need for informed consent. Additionally, no identifiers were extracted from the reports, with minimal risk or harm associated with completing the work.

## Results

### Characteristics of Canadian Red Cross (CRC) supported CHW projects in Africa

Over the last 15 years (2007–2022), CRC has supported a total of 17 projects that focused on iCCM, Reproductive Maternal Neonatal Child and Adolescent Health (RMNCAH), Sexual and Reproductive Health and Rights (SRHR), Health Systems Strengthening (HSS), and Health Promotion (HP). Over 9500 CHWs were trained through CRC supported projects and reached close to 7.5 million beneficiaries in 11 countries across Africa (Table 1). More than 70% of CRC supported projects have trained CHWs in Mali (43%) and South Sudan (27%), with the remaining in Somaliland, Kenya, Liberia, Uganda, CAR, Guinea as well as in a multi-country project that included Kenya, Mali Togo, Madagascar and Nigeria (Table 1, Fig 1).

**Table 1. Direct/Indirect beneficiaries and CHWs trained in CRC supported projects in Africa.**

| Country | # of Beneficiaries | | # of CHWs Supported by CRC | |
|---|---|---|---|---|
| Mali | 4,469,497 | 59.86% | 4,006 | 42.15% |
| Somaliland | 1,415,852 | 18.96% | 1,178 | 12.39% |
| South Sudan | 696,386 | 9.33% | 2,519 | 26.50% |
| Guinea | 444,418 | 5.95% | 225 | 2.37% |
| Liberia | 293,442 | 3.93% | 369 | 3.88% |
| Kenya | 88,030 | 1.18% | 541 | 5.69% |
| Central African Republic | 25,000 | 0.33% | 350 | 3.68% |
| Kenya, Mali, Togo, Madagascar, Nigeria | 18,291 | 0.24% | 114 | 1.20% |
| Uganda | 16,000 | 0.21% | 202 | 2.13% |
| Total | 7,466,916 | 100.00% | 9,504 | 100.00% |

CRC's roles included: (i) capacity strengthening, planning, monitoring, collaboration, coordination, and advocacy, (ii) operational, financial and logistics support, (iii) technical expertise and (iv) training (refer to Table 2 for more details).

Table 3 describes the 17 CRC supported CHW projects implemented in Africa between 2007–2022. Health services provided by CHWs included diagnosis, treatment (i.e., malaria, diarrhea, pneumonia, malnutrition, cholera, immunizations), sexual and reproductive health (i.e., family planning), services related to RMNCAH (i.e., perinatal care), common childhood diseases (i.e., iCCM), psychosocial support, HP activities and referral of complicated illnesses to health facilities. CHWs were trained in health promotion such as delivering key behaviour change communication messaging around infectious disease, nutrition (including breastfeeding), sanitation, hygiene, and importance of visiting healthcare centres. Additionally, CHWs were trained to engage their communities in risk communication, community engagement, and social mobilization (RCCE) in pandemic (COVID-19) and epidemic preparedness (i.e., Ebola Virus Disease).

## Facilitators and barriers

Data extraction of the reports revealed that different factors influenced the level and quality of volunteer engagement and successful project implementation. This included contextual conditions, organizational structure supporting program implementation, training of volunteers, participatory approaches to engaging stakeholders and communities and integration into the local ministry of health system.

With regards to ***contextual factors***, climate or political unrest impacted implementation activities, which were unique for each of the countries reports. Challenging weather conditions (heavy rainfall, floods and droughts), insecurity in conflict zones (Mali, CAR, South Sudan, Kenya) and road conditions (Kenya, Liberia) limited access to communities for volunteers and implementation teams. For example, flooding and the rainy seasons in South Sudan, Guinea and Kenya impeded access to project sites. Suggested facilitators by the Kenya project implementers included regular reporting processes regarding changing of environments/seasons and project planning around predictable climate events. International sanctions and border closures in Mali impeded access to project site, post-electoral crisis resulted in delayed activities in CAR, conflict areas resulted in fear and challenges to accessing facilities in South Sudan, and a security situation hampered activities in Kenya. Suggested facilitators included partnering with local actors to leverage resources (i.e., UNICEF) in Mali, in some cases, the target villages (Kenya) were remote with difficult terrain and high prevalence of illnesses (e.g., malaria)

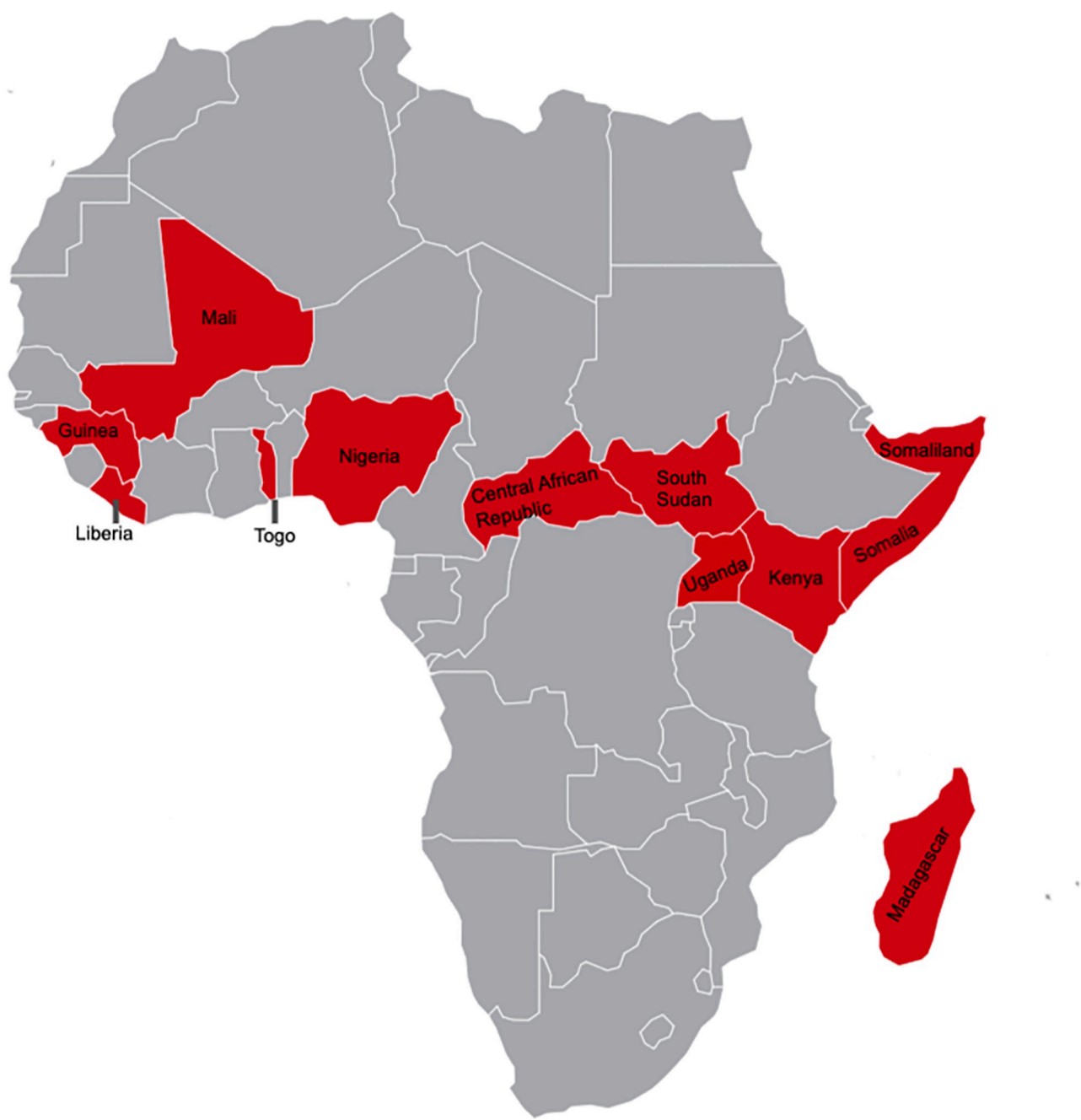

**Fig 1. Countries with Canadian Red Cross (CRC) supported Community Health Worker (CHW) programs in Africa.**

that negatively impacted volunteer engagement. In more recent projects (CAR, Somaliland), the COVID-19 pandemic challenged projected implementation due to initial unknown etiology and subsequent emergency public health orders that were implemented globally. Additionally, distrust of health care workers and the health systems/facilities (Guinea, South Sudan) in conflict affected and low resource settings was a barrier to implementation of programs that require CHW referrals to health care facilities or interaction with formal health providers.

**Table 2. Summary of the service areas and CRC role in CHW projects implemented in Africa.**

| Service areas | CRC Role |
|---|---|
| **Capacity Strengthening, planning, collaboration, coordination, and advocacy** | - Partnered with national RCRC society and Ministry of Health (MoH) and provided support planning and established and strengthened community health structure and systems to support CHWs. CRC supported national societies (NS)s to plan and facilitate consultations with local stakeholders, district health management teams, other key actors.<br>- Helped NS/MoH to prepare and establish standard operating procedures (SOPs)<br>- Partnered with other actors in advocacy for more political will and funding for community health programs (including CHWs remuneration) for sustainability.<br>- Advisory support and technical mentoring, assisting NSs in addressing issues identified with monitoring/evaluation/surveillance. |
| **Operational, Financial and Logistics support** | CRC supported the procurement of essential supplies, basic primary healthcare package and operational back up support for running the programmes lead by the NS. |
| **Training** | CRC supported NSs to train community health workers, including initial necessary trainings as well as periodic refreshers as required. Primarily, CRC's roles were to train/organize master trainers, help source training material, prepare lesson plans and provide back up support to the trainers as needed. |
| **Technical expertise** | Worked closely with local NSs to provide technical support to:<br>(i) Set up a CHWs program embedded within MoH district health systems.<br>(ii) Review the existing government health system and community health strategy and CHWs. This included training material (i.e., MoH CHW training materials or adopted existing/established material from WHO).<br>(iii) Support NS/MoH to establish community governance mechanisms (i.e., community health committees).<br>(iv) Assist with mechanisms to identify CHWs with specific consideration to gender equity<br>(v) Monitor and evaluate the programming and incorporated continuous feedback to strengthen the initiatives during the process and for future implementation/scale-up.<br>(vi) Identify the most pressing challenges facing CHW programs, such as inadequate financing, lack of supplies and commodities, low compensation of CHWs, and inadequate supportive supervision and approaches for strengthening CHW programs in different settings/countries.<br>(vii) Adapt CHW programs to mitigate additional challenges unique to humanitarian settings (armed conflicts, acute emergencies, protracted crises, displaced communities, refugees, etc.)<br>(viii) Support evidence-based practices to improve task shifting, documentation, and effectively use lessons learned for future planning.<br>(ix) Help develop monitoring and supervision tools, support NS towards rigorous monitoring, evaluation, and implementation research to enable CHW programs to continuously improve their quality and effectiveness from one phase to the other within the project.<br>(x) Help NS and MoH in use of information collected through regular monitoring mechanism and sharing info with all stakeholders.<br>(xi) Support NS and MoH to document lessons learned and to share experience with relevant stakeholders. |

**Table 3. Descriptive Table of CRC Supported CHW Projects (2007–2022).**

| Year | Country | Project Name | Objective of Project | Local National Society Supported & relationship | Canadian Red Cross Role & Sub-Service Line Area | Other key players (MoH, Community Health Orgs., NGOs, other NSs) | What did CHWs do in the program (i.e., type of health service delivery) and how was the service implemented? |
|---|---|---|---|---|---|---|---|
| 2007–2010 | Multi-Country: Kenya, Mali, Togo, Madagascar, Nigeria | Integrated Child Survival | To integrate long-lasting insecticide treated bed nets (LLNs) into supplementary immunization campaigns and have LLNs distributed to caretakers of children attending the vaccination/distribution posts, ensuring post campaign "hang-up" and "keep-up" activities, implementing community based front line malaria diagnosis and treatment and providing funds of last report for supplementary immunisation activities. | Kenya Red Cross Society (KRCS), Togo Red Cross (TRC) | Operational, financial and logistics support; technical monitoring/advisory capacity strengthening, planning, collaboration, and coordination; technical expertise; training | Ministry of Health and Sanitation, IFRC | CHWs were trained and provided LLNs to communities and carried out awareness campaigns on "hang-up" and "keep-up" campaigns. Additionally, CHWs were trained and provided diagnosis, referral and treatment of uncomplicated Malaria. 1.8M LLNs were distributed in Mali, 491,000 in Madagascar, 830,000 in Togo, 560,000 in Nigeria. 3.8M in total were distributed. In Togo, CHWs also undertook a nutrition campaign and in Nigeria a measles campaign. CHWs achieved a 92% fever treatment for children within 24 hours. |
| 2011–2012 | South Sudan | Warrap Integrated Community Health | To help meet the basic needs of conflict and natural disaster-affected communities in Warrap State, particularly women & children; key objectives include improved access to appropriate medical treatments and services; increased use of improved water supply and sanitation; improved coordination among communities, MoH, RWSS and SRCS in addressing communities' health, water and sanitation priorities). | South Sudan Red Cross (SSRC) | Capacity strengthening, recruitment, training, financial and logistic supports (boreholes), technical expertise (iCCM and Health Promotion) | Ministry of Health; Ministry responsible for water at the time in South Sudan | CHWs were trained in Community based health and first aid (CBHFA), delivered health messages/care for: malaria, diarrhea, pneumonia, polio, malnutrition, breastfeeding, newborn care, water purification/contamination |
| 2011–2015 | South Sudan | Building Community Resilience Project Eastern Equatoria, South Sudan | To increase the food security and sustainable farming practices of communities in eight Bomas in Korimi and Lotukei Payams, Budi County, Eastern Equatoria, South Sudan. | South Sudan Red Cross (SSRC) | Capacity strengthening, recruitment & training of Youth, financial support (Health Systems Strengthening & Health Promotion) | The South Sudan Ministries of Agriculture and Forestry, Water, and Health; Department of Foreign Affairs and International Development of the Government of Canada (DFATD) (Canada) | Community members received training on proper gardening techniques (11 Farmer Schools created to learn new agricultural practices). Project supported rehabilitation of 16 boreholes and addition of 11 new ones for access to clean water. Trained communities worked as pump mechanics/repairers. Youth volunteers were recruited and trained to deliver BCC messaging on sanitation, nutrition, hygiene in their communities. Youth also received vocational skills to improve business acumen and trained in humanitarian principles and conflict resolution. |

*(Continued)*

**Table 3.** (Continued)

| Year | Country | Project Name | Objective of Project | Local National Society Supported & relationship | Canadian Red Cross Role & Sub-Service Line Area | Other key players (MoH, Community Health Orgs., NGOs, other NSs) | What did CHWs do in the program (i.e., type of health service delivery) and how was the service implemented? |
|---|---|---|---|---|---|---|---|
| 2011–2015 | Mali | Improving Maternal, Newborn and Child Health in Mali | Aimed to increase efforts to reduce neonatal, under-5 and maternal mortality through the scaling up of iCCM and MNCH services at the village level by Community Health Workers (CHWs), and well-trained CRM volunteers, supported by Village Health Committees. (1) increased use of MNCH services and preventative practices at the community level by women, male and female children under-five (2) improved quality, including gender sensitivity, of MoH MNCH services at the regional, district and community levels. | Croix-Rouge malienne (Malian Red Cross) | Operational, financial and logistics support; capacity strengthening, planning, collaboration, and coordination; technical expertise; training | Ministry of Health, Fédération Nationale des Association de Santé Communautaire (National Federation of Community Health Associations) | CHWs were trained to assess, treat, and refer cases of malaria, pneumonia, and diarrhea in children under 5 (iCCM). The specific services provided included health education, promotion of prenatal and postnatal visits, provision of family planning methods, household visits, and diagnosis and treatment of common childhood illnesses. They also promoted the community referral pathway and the inclusion of men in women's health. |
| 2012–2015 | Kenya | Improvement of Resiliency in Communities Due to Chronic Drought Conditions in West & East Pokot, Kenya | To enhance food security, reducing water borne diseases, promoting better sanitation practices, and increasing resilience to common livelihood and health risks in the area. Worked on improved resilience to hazards and shocks in 11 targeted communities by increasing community capacity to mitigate risks and support them diversifying their livelihoods through grassroots approaches. | Kenya Red Cross Society (KRCS) | Capacity strengthening, recruitment, training, financial and logistic supports (boreholes), technical expertise (Health Systems Strengthening & Health Promotion in context Climate Change & Health) | UNICEF | KRCS established and trained 60 hygiene promotion CHWs to promote behaviour change including: community health training sessions; public health education campaigns; school-based hygiene education; and household visits by trained volunteers to deliver health education and services. |
| 2012–2015 | Kenya | Improving Maternal, Newborn and Child survival in West Pokot County, Pokot Central district, Kenya | To work with communities and relevant Ministries to scale-up delivery of basic health services to poor people living in rural areas. This part of the project focuses on maternal child health services and rolling out iCCM approach through CHWs as primary-level intervention in the hopes MoH-Community lineages will be strengthened. | Kenya Red Cross Society (KRCS) | Technical support (project management, monitoring/evaluation), technical monitoring/ advisory capacity strengthening, planning, collaboration and coordination (RMNCAH) | Agency for Technical Cooperation and Development (ACTED) and Ministry of Health in Kenya | Trained CHWs received reporting tools, information Education Communication (IEC) materials, drugs and equipment to facilitate health education and management and referral of cases of sick children (had bicycles to transport themselves to respective communities). CHWs managed diarrhea cases with ORS and Zinc and continued referring cases <5 with pneumonia and severe cases of malaria to health facilities. Through ACETED, communities were supported on income generation. |

*(Continued)*

**Table 3.** (Continued)

| Year | Country | Project Name | Objective of Project | Local National Society Supported & relationship | Canadian Red Cross Role & Sub-Service Line Area | Other key players (MoH, Community Health Orgs., NGOs, other NSs) | What did CHWs do in the program (i.e., type of health service delivery) and how was the service implemented? |
|---|---|---|---|---|---|---|---|
| 2012–2015 | Liberia | Improving Maternal, Newborn, and Child Survival in Liberia | To reduce maternal, neonatal, and child morbidity and mortality rates in Liberia by improving access to and quality of MNCH services. Specifically, the project aimed to improve the primary health care services through a standard set of outreach, preventive, promotion, basic curative and referral services. | Liberia National Red Cross Society (LNRCS) | Technical and project management support from the Canadian Red Cross (RMNCAH) | Ministry of Health and Social Welfare (MOHSW) in Liberia | Community health volunteers were trained to assess, treat, and refer cases of malaria, pneumonia, and diarrhea in children under 5. The specific services provided by volunteers included health education, counseling, provision of family planning methods, and diagnosis and treatment of common childhood illnesses. They provided these services through a door-to-door approach in their communities, and they were supervised by community health officers (CHOs) and nurses. They used no touch technique which was promoted by WHO/MoH during the outbreak of Ebola virus in 2014. The program also provided the volunteers with bicycles and other supplies to facilitate their work. |
| 2013–2015 | Somaliland | Community Resilience Project | To enhance resilience of target communities as well as their capacity to effectively respond to disasters through (i) Disaster Management (DM) capacity strengthening of SRCS—training staff and volunteers on disaster risk reduction (DRR) and climate change adaptation (CCA); (ii) increasing community knowledge and awareness on natural hazard/climate change—recruiting community social mobilizers, (iii) improving hygiene practices for safe water & (iv) strengthening community livelihoods through small scale DRR initiatives (support farmers with ploughing ahead of rainy seasons, alternative livelihoods. | Somali Red Crescent Society (SRCS) | Operational, financial and logistics support (Health Systems Strengthening, Health Promotion in context of Climate Change & Health) | German Red Cross; Norwegian Red Cross; informal partnership with Ministry of Agriculture, Environment and Water Resources in Somaliland and Amu University | Volunteers were trained in community emergency response teams across six regions. Additionally, Government staff members and volunteers received training on basic disaster risk reduction (DRR) and climate change adaptation (CAA) to enhance community knowledge and awareness regarding natural hazards and climate change. To facilitate effective communication and information dissemination within these communities, six community social mobilizers were recruited and trained to conduct information, education, and communication (IEC) activities to actively involve and inform their respective communities. |

(*Continued*)

**Table 3.** (Continued)

| Year | Country | Project Name | Objective of Project | Local National Society Supported & relationship | Canadian Red Cross Role & Sub-Service Line Area | Other key players (MoH, Community Health Orgs., NGOs, other NSs) | What did CHWs do in the program (i.e., type of health service delivery) and how was the service implemented? |
|---|---|---|---|---|---|---|---|
| 2014–2018 | Liberia | The Ebola Case Management in West Africa project | To improve the recovery process and reduce the vulnerability of people affected by the Ebola Virus Disease (EVD) by strengthening and promoting the use of essential health services; improving access to water, sanitation and hygiene (WASH) in targeted health facilities and schools; improving psychosocial support (PSS) for Ebola-affected communities; and enhancing LNRCS and communities' preparedness and response to emergencies and epidemics. | Liberia National Red Cross Society (LNRCS) | Operational, financial and logistics support (Health Systems Strengthening and Health Promotion) | Ministry of Health and Social Welfare (MOHSW) in Liberia, NGOs including UNICEF, WHO, PLAN, MEALS, ACF, DRC | Volunteers were trained and carried out Community Based Health and First Aid (CBHFA), Community Event Based Surveillance (CEBS), epidemic control for volunteers, and logistics for health disaster response for LNRCS chapter. They also provided Information, Education, Communication (IEC) / Behaviour Change Communication (BCC) materials on epidemics to communities (e.g.: EVD, Lassa fever, bloody diarrheas, cholera and Ebola). They also conducted awareness activities with targeted communities to promote the re-utilization of health facility services especially in the areas of Reproductive, Maternal, Newborn and Child (RMNC) health (including vaccination campaigns), including WASH/Hygiene awareness sessions in schools. Were trained to Identify, assess and refer cases requiring special PSS and report info to the national referral system for specialized care (mental health services, child services, child protection measures). Facilitated the provision of direct psychological first aid to individuals and families through peer-to-peer approach (women and girl friendly spaces). |

(*Continued*)

**Table 3.** (Continued)

| Year | Country | Project Name | Objective of Project | Local National Society Supported & relationship | Canadian Red Cross Role & Sub-Service Line Area | Other key players (MoH, Community Health Orgs., NGOs, other NSs) | What did CHWs do in the program (i.e., type of health service delivery) and how was the service implemented? |
|---|---|---|---|---|---|---|---|
| 2014–2018 | Guinea | The Red Cross and Ebola Recovery in Guinea: | To improve the quality of the services in the sanitary structures, with particular attention to the prevention and control of infections (special focus on Ebola but also meningitis, cholera, malaria), improve community capacity to detect, treat and isolate cases, community awareness activities and the reduction of burdens imposed on households to access health services and medicines including raising confidence of the healthy system among populations. | Guinea Red Cross (GRC) | Capacity strengthening, planning collaboration and coordination as well as operational, financial, logistics and technical support. CRC participated in the emergency health response and guide the Guinean Republic through their Strengthening of the local health system (Health Systems Strengthening and Health Promotion). | French Red Cross (FRC); Ministry of Health and Public Hygiene (at the national, regional, and prefectural/district levels); Regional Health Directorates, Local Ministry of Health; Health Facilities in Macenta and Guéckédou | Volunteers participated in community raising initiatives on health indicators, symptoms, treatment, prevention, and isolation requirements of infections including Ebola, importance of visiting healthcare centres, proper hygiene, psycho-social support, and personal protective equipment. Red cross volunteers trained and provided supportive supervision to health facility staff to support skills/practice which included immunizations, raising awareness of common MNCH conditions (i.e., child diarrhea, pneumonia) and epidemics (i.e., Ebola). Volunteers also provided bridging between health facilities and communities to provide community-based services. About 14 health facilities were rehabilitated with solar panels to help power water, electricity, sanitation. Staff were trained Training of healthcare staff in facilities on the maintenance of proper water quality and waste removal, and hygiene measures. |

(*Continued*)

**Table 3.** (Continued)

| Year | Country | Project Name | Objective of Project | Local National Society Supported & relationship | Canadian Red Cross Role & Sub-Service Line Area | Other key players (MoH, Community Health Orgs., NGOs, other NSs) | What did CHWs do in the program (i.e., type of health service delivery) and how was the service implemented? |
|---|---|---|---|---|---|---|---|
| 2014–2019 | South Sudan | "Improving Maternal, Newborn and Child Survival Project in the Gogrial West, Warrap State, South Sudan | To reduce maternal, newborn and child mortality by increasing the awareness of, demand for, and access to a range of health services at the community level, including Integrated Community Case Management of childhood illnesses (iCCM); health promotion and prevention; Water, Sanitation and Hygiene (WASH); and capacity strengthening in a range of areas. South Sudan Red Cross. | South Sudan Red Cross (SSRC) | Operational, Financial and Logistics Support (RMNCAH) | Global Affairs Canada (GAC) & Ministry of Health in South Sudan | The project implemented the newly established Boma Health Initiative, which is the Ministry of Health's community based-health strategy. CHWS were trained and delivered health promotion messages in communities as well as resources (i.e., reusable sanitary pads, soaps, insecticide treated nets) delivered to women, newborns, children; they also provided iCCM (identified & treated uncomplicated Pneumonia, Malaria and diarrhoea cases); an additional component on menstrual hygiene promotion delivered by red cross volunteers was piloted in schools and led to current project for South Sudan dedicated to menstrual hygiene. Close to 70 bore holes were drilled to provide clean water to communities. Created water management community committees, so many women were trained on how to manage borehole as well as had mechanics trained to repair boreholes if broken. Water management community committees were trained by project team, and each village had two community (Home Health Promoters) who were linked with water management committees to continue awareness. For EVD messages: The aim of this approach is to reach one on one, discuss some of the misconception and myths of community in relation to Ebola Viral disease in order to the have clear understanding of the cause, signs and symptoms, transmission, and preventive measure of the disease. |

*(Continued)*

**Table 3.** (Continued)

| Year | Country | Project Name | Objective of Project | Local National Society Supported & relationship | Canadian Red Cross Role & Sub-Service Line Area | Other key players (MoH, Community Health Orgs., NGOs, other NSs) | What did CHWs do in the program (i.e., type of health service delivery) and how was the service implemented? |
|---|---|---|---|---|---|---|---|
| 2016–2021 | Mali | Maternal, Newborn and Child Health Project in Mali (2016–2021) | To contribute to the reduction of maternal and infant mortality in the regions of Sikasso and Koulikoro and covered six health districts. for improved health service delivery and utilization for mothers, newborns and children under five as well as Increased use of maternal, newborn and under-five child health data by policy makers. | Croix-Rouge malienne (Malian Red Cross) | Operational, Financial and Logistics Support; Capacity Strengthening, planning, collaboration, and coordination; Training | SickKids Hospital in Toronto; Global Affairs Canada (funding), Ministry of Health and Social Development of Mali, Pharmacie Populaire du Mali; ACTED (French NGO); MenEngage | The project supported 440 CHWs and 160 primary health clinics in Koulikoro Region. CHW were trained in and delivered iCCM: assessment, diagnosis, and treatment of uncomplicated cases of malaria, pneumonia, cold/cough and diarrhea for children <5 yrs. and referring complicated cases. Screening and referral of newborns with danger signs. Awareness and counselling on family planning and distribution of select modern contraceptives. Screening and referral of children under 5 for malnutrition. Engaged different community stakeholders, including religious leaders and village chiefs, in discussions & activities about social norms that undermine MNCH. Operational research in using low-does high-frequency training to facility staff; Supported select community structures (ASACOs) generate income to fund their clinics and CHWS. Installation of solar energy infrastructure in rural maternity homes to enable delivery services 24 hours a day. |
| 2018–2019 | Uganda | Integrated Community Health and Epidemic Readiness | The purpose of the project is two-fold: 1) to improve the health of South Sudanese refugees and host communities in the West Nile region. This is done by increasing the capacity of URCS volunteers in providing community-based health services with a focus of women and children's health, as they are the most vulnerable groups, as well as increasing capacity of URCS volunteers to prevent, detect and respond to potential disease outbreaks in the targeted communities; 2) to improve the psychological well-being of South Sudanese refugees and host communities in the West Nile region. | Ugandan Red Cross Society (URCS) | Health and SGBV-related technical support as well as operational and logistics support (project management). | Icelandic Red Cross (IceRC); Refugee and host communities at Bidi Bidi, Imvepi and Rhino settlements, Government of Uganda, UNHCR and other UN agencies such as UNICEF, WFP and UNFPA, RCRC partners, IFRC and ICRC and Non-Governmental Organizations such as OXFAM, AIRD, MSF, World Vision and others. | Increased the capacity and knowledge of URCS volunteers in SGBV prevention practices to conduct community outreach activities. These activities complemented the capacity strengthening of URCS volunteers in Psychosocial Supports (PSS) to support targeted communities in responding to their mental health needs. |

(Continued)

**Table 3.** (Continued)

| Year | Country | Project Name | Objective of Project | Local National Society Supported & relationship | Canadian Red Cross Role & Sub-Service Line Area | Other key players (MoH, Community Health Orgs., NGOs, other NSs) | What did CHWs do in the program (i.e., type of health service delivery) and how was the service implemented? |
|---|---|---|---|---|---|---|---|
| 2019–2020 | South Sudan | Ebola Virus Disease (EVD) Preparedness for South Sudan | To increase Ebola Virus Disease Preparedness and Response in South Sudan. | South Sudan Red Cross (SSRC) | Collaboration/ Coordination; Technical expertise; Training (Health Systems Strengthening and Health Promotion) | The EVD National Task Force (NTF); IFRC; Ministry of Health (MoH) and Government of South Sudan, EVD National Task Force (NTF), WHO + other NSs = Danish Red Cross, Netherlands Red Cross, Swedish Red Cross, Norwegian Red Cross, Finish Red Cross, Norwegian Red Cross, Swiss Red Cross, Canadian Red Cross and Turkish Red Crescent. | Volunteers were trained (32 females and 46 males) on Safe and Dignified Burial (SDB) in the two locations (Nimule and Yambio) and on Risk Communication, Community Engagement and Social Mobilization (RCCE). CHW's worked with their communities to discuss some of the misconception and myths of community in relation to Ebola Viral disease in order to the have clear understanding of the cause, signs and symptoms, transmission and preventive measure of the disease. Safe and Dignified Burial was effective in preventing the spread of the infection when endorsed locally by community leaders through a by-law. The CHW team equally comprised of men and women, were well trained and sensitive to cultural aspects of the burials, and religious norms. EVD Burial sites were provided with engagement of and by the communities of in the project sites. |
| 2019-ongoing | South Sudan | Advance Partnerships in Health–RMNCAH Project in South Sudan | To increase knowledge and change behavior regarding reproductive, maternal, newborn, child and adolescent health (RMNCAH) in the community; to increase awareness of the PHC services, early referral and monitoring key critical cases after treatment at PHC units; documentation and learning from implementation to build a community health model in the context of insecurity and conflict in South Sudan. | South Sudan Red Cross (SSRC) | Capacity Strengthening, planning, collaboration, and coordination, focusing on partnerships. (RMNCAH) | International Committee of the Red Cross (ICRC) was an implementing partner. MoH was directly involved in recruitment and training of Boma Health Workers | Assisting with purchase of medical supplies for health facility, as well as training CHWs for(i) community-based case finding and referral of common illnesses of childhood (malaria, diarrhea, pneumonia) as well as antenatal care, danger signs, delivery, and family planning, (ii) defaulter tracing for immunization (iii) nutritional screening and referral on RMNCAH topics as well as (iv) health and hygiene promotion on several RMNCAH topics safe (safe motherhood practices, nutrition, child health, family planning, sexually transmitted diseases, sexual and gender-based violence including referral channels, psychological first aid,). |

*(Continued)*

**Table 3.** (Continued)

| Year | Country | Project Name | Objective of Project | Local National Society Supported & relationship | Canadian Red Cross Role & Sub-Service Line Area | Other key players (MoH, Community Health Orgs., NGOs, other NSs) | What did CHWs do in the program (i.e., type of health service delivery) and how was the service implemented? |
|---|---|---|---|---|---|---|---|
| 2021–2022 | Somaliland | Strengthening Service Delivery of Emergency Response in Somaliland | To improve the response of the Somali Red Crescent Society (SRCS) to the health and emergency needs of vulnerable populations affected by the COVID-19 crisis, the pre-existing complex humanitarian situation, and natural disasters in Somaliland. To address the health in emergency needs, the project supported mobile health clinic operations to provide integrated health and nutrition services to most remote communities during the second and third waves of COVID-19. The project also supported the implementation of the SRCS COVID-19 Plan of Action, in coordination with other Partner National Societies (PNS), to combat the second and third waves of COVID-19 and build vaccination awareness. The project built institutional capacity on epidemic preparedness and response (EPR) for volunteers for a multiplying effect in communities. | Somaliland Red Crescent Society (SRCS) | Operational, Financial, Logistics and Technical support. | Ministry of Health Development (MoHD), Ministry of Family Planning and Family Affairs, and the Ministry of Planning and Development of Somaliland, Icelandic Red Cross | During the project period SRCS G&G, EPR and PHC Managers carried out a series of trainings on the PGI component. These trainings covered PSEA, SGBV-VAC, FGM, and PSS for SRCS staff, volunteers and PHC staff. SRCS trained 126 SRCS staff, 688 volunteers and 19 PHC staff on PGI. |

(*Continued*)

**Table 3.** (Continued)

| Year | Country | Project Name | Objective of Project | Local National Society Supported & relationship | Canadian Red Cross Role & Sub-Service Line Area | Other key players (MoH, Community Health Orgs., NGOs, other NSs) | What did CHWs do in the program (i.e., type of health service delivery) and how was the service implemented? |
|---|---|---|---|---|---|---|---|
| 2020-Ongoing | Central African Republic | Advance Partnerships in Health–RMNCAH Project in CAR | To increase knowledge and change behavior regarding reproductive, maternal, newborn, child and adolescent health (RMNCAH) in the community; to increase awareness of the PHC services, early referral and monitoring key critical cases after treatment at PHC units; documentation and learning from implementation to build a community health model in the context of insecurity and conflict in Central African Republic. | Central African Republic Red Cross Society (CARCS) | Operational, Financial, Logistics and Technical support. Recruitment and training of CHWs and equipment with outreach materials (bicycles, backpacks), support CAR NS with community engagement. | Ministry of Health in CAR | Training CHWs for (i) community-based case finding and referral of common illnesses of childhood (malaria, diarrhea, pneumonia) as well as antenatal care, danger signs, delivery, and family planning, (ii) defaulter tracing for immunization (iii) nutritional screening and referral on RMNCAH topics as well as (iv) health and hygiene promotion on several RMNCAH topics (safe motherhood practices, nutrition, child health, family planning, sexually transmitted diseases, sexual and gender-based violence including referral channels, psychological first aid,). Additionally, through community engagement, outcomes included (i) a community-based referral system where communities themselves identified what type of system and support they need to have a referral system (i.e., who need to go to health facility), so the community led and owned initiative included 6 community motorbike ambulances to transport women in labour, (ii) as a result of the CHW sensitization work, 2500 new latrines and 1500 dish racks were constructed by the community themselves (their own initiatives). |

Refer to Supplement 1 for a detailed outline of each country contextual factor barrier and facilitator.

On a ***structural level*** the lack of organizational structure to support volunteers, lack of resources (Guinea, Liberia) to run an office and proposed program, delayed payments, lack of commodities/materials (i.e., medicines, contraception, mosquito nets) and IT infrastructure (I.e., internet, hardware) limited volunteer engagement. Not only did it impact training of CHWs, but they were also unable to engage their communities and provide the key services they were trained for (i.e., family planning, iCCM, communication materials for health messaging). Alignment with the ministry of health (MoH) strategy (Mali, Kenya), ensuring that volunteers are known by health authorities, being equipped to host volunteers or the presence of a stable and reliable team, the provision of incentives (e.g., honoraria/renumeration, bikes

and mobile phones) (Liberia) as well as recruitment of volunteers through the national society encouraged engagement in volunteerism and, by extension, the community (CAR, Mali, Kenya).

*Regular training as well as refresher training* and the incorporation of learning in the activities was identified as an important factor to increase the capacity of volunteers and enhance knowledge and skills to provide quality health services. One report also mentioned the importance of providing certificates (Liberia). Barriers relating to the uptake of training were a low literacy level and challenges around retainment of information (Liberia, Kenya). In one context (South Sudan, 2014–2019), implementers replaced the training-of-trainers approach with more direct training in each community because they found that knowledge transfer occurred more commonly through "horizontal" family connections.

*Participatory approaches* that contributed to involving and creating an interest or strong will in communities, facilitated the work of volunteers and helped to ensure continuation of the project Liberia, South Sudan, CAR). In Kenya it was noted that the engagement of women and youth and the ability of women to take on leadership positions influenced the community positively. In South Sudan, during a dignified burial process, community leaders were able to mobilize youth volunteers to dig the grave. At the same time, it was also noted that distrust in the community (Guinea), low visibility of the CHWs and challenges around inadequate community engagement negatively impacted the volunteers' activities (Liberia, Mali).

The *integration of CHWs (and programs related to CHWs) into the local Ministry of Health (MoH), or health infrastructure*, varied across the different countries depending on the context, objectives, and approaches of each project. The analysis also identified some factors that made integration more successful or more challenging. The projects that focused on MNCH tended to have a closer and more collaborative relationship with the local MoHs than the projects that addressed other health issues, such as Ebola or drought. This may be because MNCH is a priority area for most MoHs and requires a strong health system and referral network. In the case of the South Sudan MNCH project, CRC engaged the MoH from project onset by working with their staff as master trainers for Boma health workers (South Sudan 2014–2019). The projects that involved multiple partners, such as community health organizations, NGOs, NSs, or ICRC, had to coordinate and align their activities and expectations with the local MoHs and each other. The projects that adapted to the different political, social, cultural, and environmental contexts of each country and community were more likely to integrate successfully with the local MoHs and other stakeholders. This involved engaging with the communities and local stakeholders in the design, implementation, and evaluation of the projects, as well as respecting and incorporating their needs, preferences, and values. Additionally, the projects that supported the training and supervision of CHWs who provided essential MNCH services at their community level were able to bridge the gap between formal health facilities and community members, especially in remote or underserved areas. However, they also noted facing challenges in ensuring the quality, motivation, and retention of CHWs, as well as their recognition and remuneration by the local MoHs.

Some key examples of integration in the projects include:

- Participatory involvement of community members in the projects—in Liberia, community health committees (CHCs) were established, and they were involved in identifying community health priorities, planning and participating in community health actions, mobilizing resources, coordinating CHW activities, and organizing community health days. By involving community leaders and members throughout the project, local integration was strengthened, and this facilitated long-term sustainability of the project.

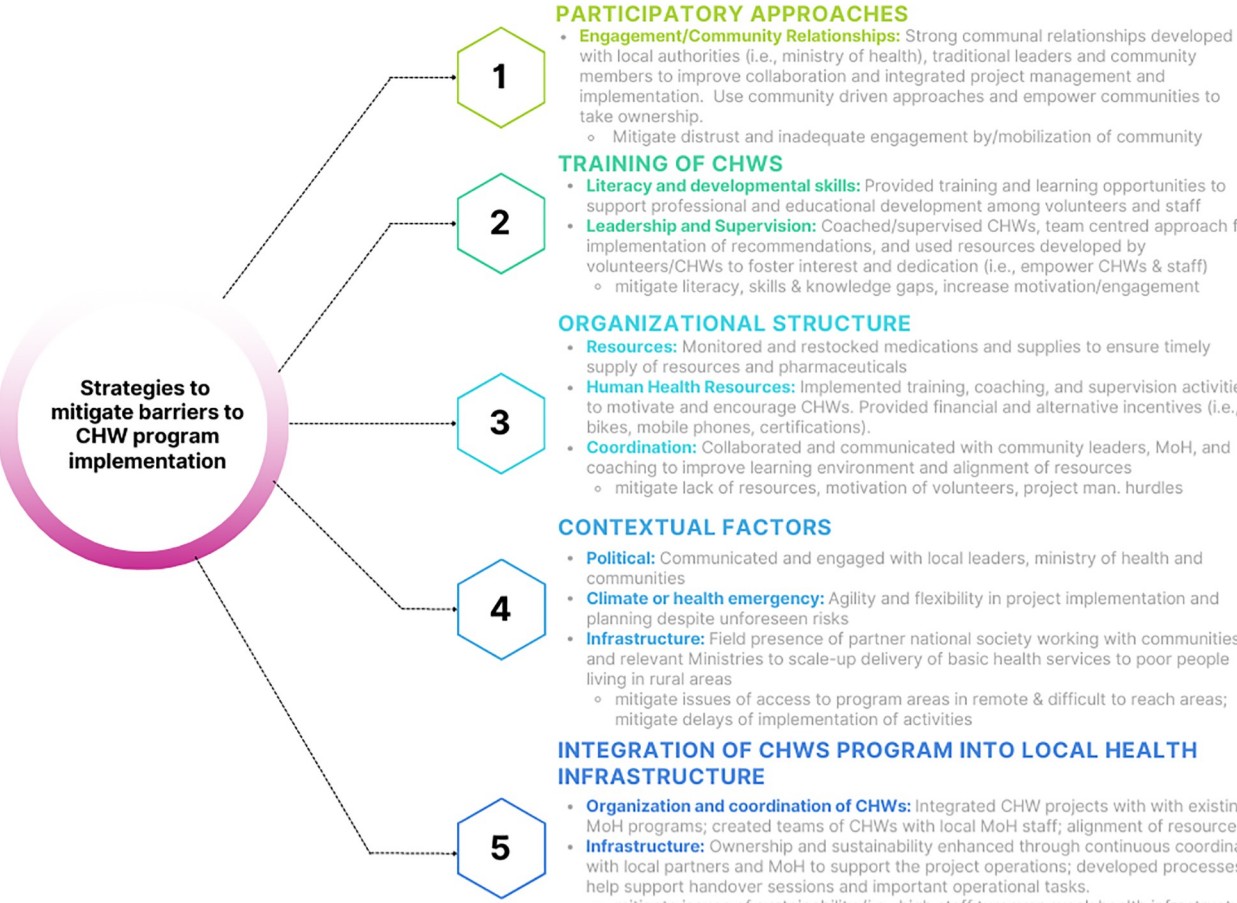

**Fig 2. Strategies to mitigate barriers and facilitators to CHW program implementation.**

- Integration of CHWs into the local health care system–in South Sudan, Somaliland, and CAR, direct referral of cases to primary health units was done within the projects. This facilitated the continuum of care and integration into existing MoH infrastructures.

- Partnerships with MOH and local health authorities—involving the local MOH in the initial planning phases of the projects is integral to implementation.

- Alignment of the program planning with the country's national priorities.

- Strategic partnerships for technical expertise–In Somaliland, partnerships with local universities facilitated the collection of data and profiling of the SRCS. The involvement of academic institutions played a crucial role in fostering local knowledge creation.

Fig 2 summarizes the findings and S1 Fig provides details extracted from each project.

## Lessons learned in CHW project implementation

The key lessons learned from CRC supported CHW programmes can be categorized into six main themes: importance of project management for proper implementation, the need for local partnership and community engagement, essential training, the value of evidence-based

research, adaptability and flexibility in programming and fundamental integration of protection, gender, and inclusion (PGI).

**1. Importance of project management for proper implementation.** Project Management includes fundamental organizational structures that outline memorandums of understanding, workplans for program implementation, logic models clearly stating key activities and related outcomes, as well as the plan for an exit strategy from the onset. **Several projects outlined the need for appropriate/realistic timelines that allow flexibility** as many of the programs implemented in conflict or low resource settings require continuous re-adjustments. Another example is rainy seasons, which cut-off routes to remote regions. Knowing and considering weather patterns when planning certain activities can mitigate logistical interference of commodity/health service delivery. **One suggested approach was holding regular review and planning meetings to support the activities of the staff implementing the program** (I. e., internal communications) as well as external communications with other actors or partners (i.e., partner organizations, funders, community health committees, CHWs, local NS). Consequently, project implementers also discussed the need to build and maintain trust among all actors involved. Planning for closure and transition strategies at the onset of the program rather than at the middle or end of project implementation was continuously emphasized, along with the need to ensure the integration of local knowledge and context, developed in a participatory manner with local actors.

More specifically to CHW training and subsequent activities, there was an emphasis on ensuring clear training, the provision of appropriate tools that CHWs can use to create health awareness/promotion, refresher schedules, and active attempts to ensure a clear supply chain for essential commodities (i.e., accounting for conflict, weather patterns) so they are well equipped to deliver services. Communication of funding was mentioned, not only for program partners implementing the project, but also for the CHWs themselves in terms of understanding the renumeration. Clear financial mechanisms, human resource policies and appropriate merit-based renumeration for CHWs, if possible, should be considered from the onset in project planning. The importance of renumeration was emphasized in several projects that outlined CHW attrition as an issue without incentives and which subsequently diminished motivation. Strategies to mitigate attrition are needed when renumeration or awards are not feasible to ensure retention of skilled volunteers.

**2. The need for stakeholder partnerships and community engagement.** Community involvement and engagement was a recurring theme across several projects outlining the need to engage with leaders, local authorities, and volunteers. This kind of engagement helped improve considerations with regards to the sociocultural context. One example was engaging religious leaders to help foster gender positive attitudes and reduce gender-based violence (GBV) through project MenEngage. Engagement also helped mitigate mistrust of international organizations, while village-level participation was deemed essential to ensure ownership and sustainability of CHW initiatives. In some contexts, teams engaged with armed forces where continued communication contributed to mitigating issues of transportation and ability to enter conflict zones.

The CRC-ICRC APiH CAR and South Sudan projects highlighted the importance of recognizing the significance of input from youth in sexual and reproductive health and rights (SRHR) initiatives; youth expressed dissatisfaction with the process, a perceived lack of acknowledgment of the importance of their input and exclusion from the project.

The importance of taking the time to build partnership and networking with partners and actors (i.e., local and national governments, partner national societies, national and international NGO's) was highlighted throughout the projects studied. Collaborative processes were thought to have helped improve linkages between MoH, National Red Cross/Crescent

societies, IFRC, CRC and other stakeholders. In particular, the importance of integration of activities, if and where possible, with local ministries and health infrastructures was empha-sized (described in more detail above).

**3. Essential training for CHWs.**   Table 4 summarizes the lessons learned regarding CHW training modalities, approaches, and subject matter content.

**4. Evidence-based approach.**   All 17 included projects demonstrated impact on outcomes through baseline and endline quantitative and qualitative evaluation approaches. Ten of the seventeen projects engaged in regular monitoring activities (quarterly or monthly depending on the program), while the reports of the remining seven did not specifically document moni-toring in final documents. Findings from monitoring data were used to continuously inform the adjustment or adaptation of the implementation activities. Establishing strong monitoring and evaluation processes that yield reliable data are the foundation for good research and con-tribute to learning and quality improvement.

Several of the extracted projects outlined the value of applying evidence-based research findings to optimize the impact of their program. The community resilience project in Somali-land (2013–2015) demonstrated how the application of appropriate evidence–based assess-ments and prioritization of CHW programs ensure targeting of communities with the highest health disparities and burden of disease. The 2007–2010 multi-country Integrated Child Sur-vival project described the need to factor in household distances and village population size as determinants for number of CHWs to train, rather than estimations that do not consider key factors highlighted through evidence.

Published findings from the implementation of CRC supported projects in Liberia, Mali, Kenya, CAR and South Sudan [10–12,23] provided valuable insights into effective practices and contributed to advancing knowledge in the provision of primary healthcare in humanitar-ian and low-resource settings. Key findings specifically related to CHW activities extracted from the project are included below:

- Behaviour changes among trained CHWs cascaded to behaviour change among community members.

- CHW activities contributed to increased number of women attending ANC visits, commu-nity access to malaria services, and reduced workload in health facilities.

**Table 4. Lessons learned regarding CHW training modalities and approaches.**

| Training modality | Approach | Subject Matter content |
|---|---|---|
| Use simulations (I.e., Ebola Virus Disease training) | Engage men in reproductive, maternal, neonatal, child, adolescent health (RMNCAH) trainings and session deliveries Engage youth in SRHR and emergency recovery (i.e., Ebola Virus disease) | Behaviour change engagement of CHW important to disseminating behaviour change material to community. |
| Use of short and pictorial messages or visual aids (i.e., for CHWs and beneficiaries with low literacy) | Include youth, elderly and people with disability to engage as volunteers | Health promotion as part of integrated community case management (iCCM) is important, particularly when incorporating into government policy/integration into MoH. |
| Training and supervision are key to success of CHW programs | Provide package of activities for training rather than separate (i.e., integration into other programs). Examples: Latrines and water points rehabilitation, construction, provision of hygiene kits and WASH awareness sessions should be organized together (as package) and earlier in the project for better impact. | |
| Ensure refresher trainings (build into the program from the onset) | Trainings should be localized (i.e., sometimes top-down training such as train the trainer may not work; rather family member to family member or within community may work better). | |

- CHW provided services in ways that are well accepted by and meet the needs of the community.

- CHW activities improved demand and acceptance for integrated community care outreach, reduction of maternal neonatal child mortality, and increased population coverage of community-based healthcare.

**5. Fundamental Integration of Protection, Gender and Inclusion (PGI) for CHW program implementation.** As outlined in Fig 3, protection, gender, and inclusion (PGI) was considered or integrated at the policy l (n = 3) and project (n = 15) level. In some cases, PGI was discussed as a gap, recommendation, lesson learned (n = 2) or not mentioned in available reports (n = 2). At the policy level, for example, an early assessment in the 2016–2022 Mali project, identified the need to support women's participation as decision-makers in the Malian Red Cross (CRM) response at branch and national levels. Jointly with CRC and other national Red Cross societies (Danish, Luxembourgish, Dutch, Belgian, French, Spanish), CRM revised the PGI national action plan to be implemented in this and future projects. In Kenya, the Ministry of Health (MoH)'s 2000 National Gender and Development Policy Framework requires a minimum presentation of 33% female CHW in projects. This was integrated into the 2012–2015 Kenya project for improving maternal newborn and child survival in an effort to promote

| Country | Year | Project Name | PGI at policy level – well integrated | PGI at program level - well integrated | PGI at program level - partially integrated | Discussed as a gap or recommendation or lesson learned | Not mentioned in current & available reports |
|---|---|---|---|---|---|---|---|
| Kenya, Mali, Togo, Madagascar, Nigeria | 2007-2010 | Integrated Child Survival | | ■ | | | |
| South Sudan | 2011-2012 | Warrap Integrated Community Health | | | ■ | ■ | |
| Mali | 2011-2015 | Improving Maternal, Newborn and Child Health in Mali | | ■ | | | |
| South Sudan | 2011-2015 | Building Community Resilience Project Eastern Equatoria, South Sudan | | ■ | | | |
| Liberia | 2012-2015 | Improving Maternal, Newborn, and Child Survival in Liberia | | ■ | | | |
| Kenya | 2012-2015 | Improving Maternal, Newborn and Child Survival Project and Improving Resiliency In communities to Chronic Drought Conditions Project in Kenya | ■ | ■ | | | |
| Kenya | 2012-2015 | Improvement of Resiliency in Communities Due to Chronic Drought Conditions in West & East Pokot, Kenya | ■ | ■ | | | |
| Somaliland | 2013-2015 | Community Resilience Project | | ■ | | | |
| Guinea | 2014-2018 | The Red Cross and Ebola Recovery in Guinea: | | ■ | | | |
| Liberia | 2014-2018 | The Ebola Case Management in West Africa project | | ■ | | | |
| South Sudan | 2014-2019 | Improving Maternal, Newborn and Child Survival Project in the Gogrial West, Warrap State, South Sudan | | ■ | | | |
| Mali | 2016-2022 | Maternal, Newborn and Child Health Project in Mali of the Canadian Red Cross (2016-2022) agents de santé communautaire (ASC) | ■ | ■ | | | |
| Uganda | 2018-2019 | Integrated Community Health and Epidemic Readiness | | | ■ | ■ | |
| South Sudan | 2019-2020 | EVD Preparedness for South Sudan | | | | | ■ |
| South Sudan | 2019-2022 | Advance Partnerships in Health – RMNCAH Project in South Sudan | | ■ | | | |
| Somaliland | 2021-2022 | Strengthening Service Delivery of Emergency Response in Somaliland | | ■ | | | |
| Central African Republic | 2020-Ongoing | Advance Partnerships in Health – RMNCAH Project in CAR | | | | | ■ |

**Fig 3. Consideration of PGI in CRC supported CHW projects.**

gender equality in the 11 communities using a participatory approach. The Kenya Red Cross Society (KRCS) deliberately required that each community health committee (CHC) be composed of a third women. While this may appear modest, the final report highlighted that it is in fact a realistic projection for the involvement of women in the Kenyan context. Project evaluation revealed that women's participation in decision-making at the community level was promoted and observed, along with equal gender representation among CHC members. This indicated significant progress in mainstreaming women's participation in communal planning processes. Women took on various committee roles, from being members to occupying leadership positions and guiding the groups. Thirteen of the projects analyzed had integrated PGI well at the program level, with an active intention to recruit women volunteers, had representation at the committees and among CHWs providing care, and considered gender-based violence approaches where applicable. PGI considerations across the projects tended to relate to increased participation and representation of women and their children. Where applicable to the program, the PGI discussion focused on children and individuals with disabilities. Interestingly, two programs–South Sudan 2011–2012 and Uganda 2018–2019 –discussed the gap in incorporating a PGI lens. In South Sudan, the main barriers discussed included entrenched cultural norms (due to the short duration addressing this was not included) and had men dominating the delivery-side of the planned activities that required volunteers. Although they attempted to recruit women, the short timelines of the project, excessive climatic and security conditions and pressure to launch did not allow for adequate time to undertake gender sensitization activities prior to volunteer selection. Similarly in Uganda, difficulties engaging women were encountered in male dominated community information sessions.

## Discussion

In this review we explored the barriers, facilitators and lessons learned from CRC supported CHW programs implemented in Africa. The review highlighted key factors and considerations relating to the implementation of CHW programs. This included the impact and potential of CHW, community engagement and collaboration, partnerships among key stakeholders, integration with ministry of health programs/policies, consideration of PGI and the importance of evidence-based approaches.

### A participatory approach

Participatory and collaborative processes were identified to have contributed to ensuring community acceptance and interest to participate as CHW/volunteers, ensure alignment of the CHW program with the MoH objectives, improve considerations with regards to sociocultural context, mitigate mistrust and improve ownership and sustainability. Some countries reported successful engagement of CHWs or volunteers, while others reported challenges around training, incentives, retainment and the lack of organizational capacity to host CHW/volunteers.

Participatory approaches are integral to CHW programming. These approaches aim to involve community members in the planning, implementation, and evaluation of projects and programs [21,24]. They include community meetings, focus group discussions, surveys, and participatory workshops. By actively engaging community members, these methods ensure that interventions are culturally relevant and responsive to local needs [24]. Additionally, it has been found that feasibility of service delivery in conflict settings depends on community inclusiveness [25]. One significant benefit of participatory approaches is the opportunity for a multicultural exchange of knowledge among participants. Such exchanges help individuals realize that they face similar challenges and enable them to share cost-effective solutions from their respective communities [24]. Recent learnings from projects implemented in CAR and

South Sudan outline the need for well-defined and clear messaging, community inclusiveness, and a localized plan for delivery of services [25]. Incorporating this aspect into future CHW-based programming can further enhance its effectiveness.

Engaging community and local actors is essential for successful implementation and sustained interest from the community [26]. Community engagement literature has shown that power imbalances must be addressed so the interest and engagement is with and from the community. This means meaningful participation that is "inclusive, accessible and supportive" of the community [26]. Alignment with partner organizations such as UNICEF, CARE, or USAID can be particularly valuable, as these organizations possess established relationships and networks within communities that can be leveraged to support community engagement efforts [27].

Regardless of the exit mechanism employed, it is crucial to ensure local alignment and an appreciation of contextual factors by the principal agents involved. This is vital for improved sustainability of CHW programs. Key decision-makers should possess an understanding of the local context while financial resources, coordination platforms, and strong relationships among stakeholders should be available [27]. The participatory approach proves beneficial in facilitating the handover of programs during exit as it encourages the continuous involvement of stakeholders throughout the program's lifespan.

As an extension of the participatory approach, engaging armed groups during conflict in order to negotiate safe passage brings forth several ethical and practical considerations beyond the scope of this document review. Not only do armed conflicts complicate engagement, but they also create dangerous situations that hinder implementation of community-based programs during emergencies [28]. Programs implemented in CAR and South Sudan revealed that community engagement, collaboration and negotiation of safe passage were key strategies to deliver services and resources in conflict settings with the support of local actors [25]. Here, we acknowledge there are additional practical considerations and several key players, such as Médecins sans frontiers (MSF) and International Committee of the Red Cross/Crescent (ICRC), have contributed important findings to inform and unpack this phenomenon [28,29]. In their work they discuss asymmetrical relationships between affected people affiliated with armed groups, armed rebels, opposing government and the humanitarian organizations attempting to provide neutral services.

## Contextual factors

In low resource, conflict affected and humanitarian settings, the implementation of community-based health programs is heavily influenced by the political context, climate conditions, emergencies, and healthcare infrastructural limitations. Political instability and conflicts can hinder access to communities, impede decision-making processes, and create additional security/safety risks for implementing community-based programs [30,31]. Health effects of climate change and associated natural disasters are not new [32]. Relatedly, in our projects, climate factors such as natural disasters and changing environmental conditions disrupted infrastructure, displaced communities, and impacted the effectiveness and sustainability of community-based initiatives. Moreover, emergency situations and the existing lack of essential infrastructure, such as reliable transportation networks and access to basic health services, posed significant challenges to the implementation and continuity of community-based programs. Mitigation strategies to address these issues include prior knowledge and continued understanding of contexts through localisation strategies. These include but are not limited to pre-assessments of relevant areas (i.e., demographics, epidemiology, seasonal patterns), particularly in conflict affected settings, as well as engagement of key actors on the ground in

political regimes, government, health system and non-governmental organizations also providing humanitarian aid. There are multiple tools and guides for assessments based on project needs (i.e., USAID Office of Conflict Management and Mitigation, health facility assessment tools, health services assessment tools) to inform strategies, policies and program design [33,34].

## Organizational structure

In almost all the studies extracted in this document review, project management and communication plans were deemed vital when implementing CHW initiatives. Effective project management is important in humanitarian settings, which operate on funding proposals and key criteria for reporting back to donors based on set budgets and resources [35]. However, they are tough to manage because of continuously shifting, dangerous and low resource settings that require agility and quick response. Proper and proactive management ensures resources are allocated efficiently, tasks are coordinated, and timelines are adhered to, facilitating successful implementation. Communication plans, both internal and external, enable clear and timely information sharing among project stakeholders, foster collaboration, address/mitigate challenges, and enhance the overall effectiveness of CHW programming being implemented. These plans help maintain coordination, ensure transparency, and promote accountability, and lead to improved healthcare delivery and better outcomes for affected communities. A key approach to support project management and implementation is monitoring and evaluation (M & E). Throughout many of the lessons learned from the extracted projects, M & E were either deemed a foundation for success or a necessity for continuous quality improvement. M & E contributes to reliable and valid data for an evidence-based approach to data-driven decisions and actions. It allows for the continuous assessment of interventions, the identification of successful strategies, mitigation of challenges and the refinement of approaches based on evidence of what works [36]. Ultimately, the M & E contributes to the generation of evidence and support learning and quality improvement in programs and interventions.

While most reports included in this review did not mention exit strategies, others emphasized the importance of designing closure strategies at the start of a humanitarian response in collaboration with different stakeholders and the unique role of CHW and volunteers in this process. Early engagement of staff, community members, leaders and key stakeholders are essential to mitigate negative consequences and to ensure longer-term positive outcomes [37]. Poorly planned project closure or handover has given rise to situations where people have reduced access to medical care, has undermined trust and has negatively influenced future humanitarian responses [38]. The importance of planning exit strategies in humanitarian settings is increasingly recognized [37,39]. As a result, various guidelines have been developed, such as those provided by the IFRC and CARE [40,41]. Despite these efforts, there continues to be lack of clarity regarding best practices and processes for implementing effective exit strategies in humanitarian contexts. Further research is needed to better understand the role of CHW and volunteers and what guidance is required to support the planning of exit strategies.

## Training

Trained CHWs and volunteers possess valuable contextual information and local practices that can contribute to a more effective implementation of health programs. In Kenya, access to prompt and effective malaria treatment was higher at endline and highest amongst the poorest and poor [12], and access to malaria treatment among children was 5.7 times higher when CHWs were the sources of care sought [11]. In a climate of distrust of healthcare and health providers, as would happen in an Ebola outbreak, CHWs were trusted by the community for

advice and disease prevention education because they understood the social and cultural complexities [10]. Extracted projects that focused on the training and supportive supervision of CHWs to deliver essential Maternal, Newborn, and Child Health (MNCH) services at the community level suggest clear contributions in bridging the gap between formal health facilities and community members particularly in remote, underserved and poor communities, as demonstrated in the published study implemented in Kenya [11].

However, challenges were identified in the quality, motivation, low-literacy and retention of CHWs, as well as their recognition and remuneration by donors and/or the local MoHs. The literature supports the importance of providing training, mentoring and regular refreshers [42] while also collecting appropriate data by CHW supervisors and external evaluators to inform and enhance CHW quality of care [23]. Although low literacy levels were identified as a barrier in several of the extracted projects, appropriate supervision and training methods can be tailored to focus on experiential learning and practical skills acquisition with an uninterrupted supply of necessary resources [43–45]. Furthermore, exploring alternative funding sources and providing incentives such as bicycles or other resources that support CHW roles and responsibilities can help sustain their motivation and dedication to the community [46].

By adopting a collective programming approach and engaging in joint efforts for resource mobilization, these projects have developed a common plan to address multiple health challenges effectively. Moreover, an integrated approach has been emphasized, incorporating CHW programs into the existing infrastructure of the local Ministry of Health (MoH) to ensure sustainability and long-term impact.

## Mainstreaming protection gender and equity

Mainstreaming and integrating PGI is needed to address (i) violence and keep people safe from harm, (ii) discrimination and people's different needs/risk, and (iii) exclusion by meaningful involvement of excluded people [47]. More specifically, PGI in training programs is essential to ensure that staff and volunteers are equipped with the knowledge and skills necessary to promote inclusivity, gender equality, and protection within their work. This involves incorporating PGI principles and standards into training curricula and methodologies, fostering a culture of respect and non-discrimination, and empowering individuals to actively contribute to addressing the unique challenges faced by different groups, including women, children, youth, persons with disabilities and older individuals.

In the lessons learned across several of the extracted studies, implementers discussed the importance of women's engagement and empowerment in CHW programs. They noted that proactive efforts should be made to ensure that women CHWs are not burdened with additional responsibilities but rather provided with opportunities for influence and decision-making. A recent article suggests that existing humanitarian culture and behaviour may be a barrier to gender mainstreaming, particularly for promoting gender equality and outline "saviour mentality and macho culture of a male-dominated field, a tolerance for abuse, and the short-term nature of crisis response" as main obstacles [48]. In contexts where male dominance prevails, it is crucial to challenge gender norms and promote equal participation and representation. By empowering women as CHWs, they can not only deliver essential care but can also influence village health committees and advocate for the delivery of care in a manner that addresses the unique needs of women and girls. That said, evidence from Ethiopia suggests that although there are new opportunities and roles offered to women who participate in CHW programs, the existing complexities of program impacts on women's lives, the reinforced gender hierarchies of work without compensation, existing domestic responsibilities, and limited ability to exercise political power or gain authority, remain [49]. Recognizing these

limitations, multi-pronged strategies could be implemented to empower women and support their leadership development. This can be achieved through mentorship programs, appropriate compensation and community engagement regarding cultural/behavioural norms (i.e., working with men and boys, engaging leaders/elders) that encourage women's active participation as healthcare providers and in the shaping of health policies and practices. Findings from CAR and South Sudan reveal that empowering women and adolescents with context-specific resources helps mitigate security, knowledge and literacy gaps [25].

While mainstreaming PGI may present challenges, organizations like the IFRC and CRC possess the expertise and capacity to support this process. Drawing on their knowledge and experience, these organizations can provide guidance, resources, and training to ensure that PGI considerations are integrated effectively into organizational policies, implementation activities, and training programs. A key resource is the Minimum standard for PGI in emergencies [50].

### Implications of findings

The global shortage of healthcare professionals underscores the critical need for CHWs to bridge the gap and provide essential healthcare services at the community level [21]. CHWs have been shown to play a vital role in delivering primary healthcare, health education, and preventive services, particularly in underserved and remote areas, as well as armed conflicts, where access to formal healthcare facilities is limited [1,2]. Findings from this work demonstrate key mechanisms integral to CHW program implementation. Understanding the importance of a localization lens, PGI integration, leveraging the strengths of local societies and their communities, and alignment with local government health priorities and directions, CRC and partner stakeholders have the expertise and experience to support CHW programming. Subsequently, the CRC—collaboratively with IFRC as well as ICRC partners—can work closely with local national societies and their governments to support implementation of CHW initiatives in conflict affected and humanitarian settings in the Africa region.

### Strengths and limitations

The limitation of this document review is that the availability and quality of documents varied in terms of consistency of information provided. There was heterogeneity in terms of types of documents available for extraction for the 17 projects as the reporting was not consistent. Additionally, the intent of reports extracted was not for research purposes to answer our research question and therefore this document review considers only available and accessible reports and does not claim generalization to all relevant reports not accessible or included. The multi-country report of Togo, Mali, Madagascar, Nigeria and Kenya was only available as an outcome for the whole project rather than by each country, which limited our ability to analyze individual country elements. Consequently, reports did not focus only on CHW/volunteers' impact or best practices, and they are based on practical experiences/reports in humanitarian settings. The strength of this study is that it is a review of facilitators, barriers and lessons learned from implemented projects, situated in the context of available research.

### Conclusions

The research study identified major outcomes, challenges, solutions, facilitators, and lessons learned from CRC supported CHW program implemented in Africa over the past 15 years. Additionally, in a post-pandemic era of workforce shortages, this work informs implementation of CHW programs which are gaining traction globally. Key findings from the review suggest that successful integration with the local health authorities was more likely when projects

were aligned with the national health strategy, involved close collaboration with the MOH and other stakeholders, considered PGI, and built on existing community structures, engagement and partnerships. Challenges to integration included weak health systems, lack of political commitment, inadequate funding, and inadequate training of health workers and CHWs.

Overall, CHWs are a critical health workforce in promoting community-based health, improving access to care, addressing health disparities, and achieving universal health coverage. They play a crucial role in healthcare systems and have a significant impact on improving health outcomes, especially in underserved, conflict affected and marginalized communities. Their unique position within communities and their ability to deliver culturally appropriate, person-centered care make them invaluable in ensuring equitable healthcare for all.

## Supporting information

**S1 Fig. Barriers and facilitators of CRC supported CHW projects.**
(PDF)

## Acknowledgments

We extend our heartfelt gratitude to the local stakeholders, particularly the Red Cross/Red Crescent National Societies, community health workers, and respective Ministries of Health (Mali, Somaliland, South Sudan, Guinea, Liberia, Kenya, Central African Republic, Togo, Madagascar, Nigeria, Uganda) whose unwavering commitment and collaboration made these projects possible across all the countries of implementation. Furthermore, we would like to express our sincere appreciation to Mr. Ashak Sherrif and the dedicated CRC Africa team for their continued encouragement and invaluable support throughout these endeavors. We are immensely thankful to Dr. Ayham Alomari, Dr. Mariam Kone, Mr. Luay Basil, and Ms. Julia Hajjar for their significant contributions in editing and proofreading the manuscript, enhancing its quality and clarity. We would also like to thank Ms. Lavinia Lakhan for her creative contributions to the barriers and facilitators S1 Fig and the manuscript Fig 2. We extend our profound appreciation to Global Affairs Canada for their global commitments to the health of women, adolescents and children, and their generous contributions across many of the projects studied in this review, and played a crucial role in the CRC supported CHW programs implemented in Africa.

## Author Contributions

**Conceptualization:** Dina Idriss-Wheeler, Mekdes Assefa, Faiza Rab, Sanni Yaya, Salim Sohani.

**Data curation:** Dina Idriss-Wheeler, Ilja Ormel, Mekdes Assefa, Christina Angelakis, Salim Sohani.

**Formal analysis:** Dina Idriss-Wheeler, Ilja Ormel, Mekdes Assefa, Christina Angelakis, Salim Sohani.

**Methodology:** Dina Idriss-Wheeler, Faiza Rab, Sanni Yaya, Salim Sohani.

**Project administration:** Dina Idriss-Wheeler.

**Supervision:** Dina Idriss-Wheeler, Salim Sohani.

**Validation:** Salim Sohani.

**Writing – original draft:** Dina Idriss-Wheeler, Mekdes Assefa, Faiza Rab.

**Writing – review & editing:** Dina Idriss-Wheeler, Ilja Ormel, Mekdes Assefa, Christina Ange-lakis, Sanni Yaya, Salim Sohani.

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
