## [Decision Letter · Decision Letter 0]

2 Oct 2023

PGPH-D-23-01553

Engaging Community Health Workers (CHWs) in Africa:

lessons from the Canadian Red Cross supported programs

Dear Dr. Idriss-Wheeler,

Thank you for submitting your manuscript to PLOS Global Public Health. After careful consideration, we feel that it has merit but does not fully meet PLOS Global Public Health’s publication criteria as it currently stands. Therefore, we invite you to submit a revised version of the manuscript that addresses the points raised during the review process.

We look forward to receiving your revised manuscript.

Kind regards,

Ama Pokuaa Fenny, Ph.D

Academic Editor

Journal Requirements:

1. Some material included in your submission may be copyrighted. According to PLOS’s copyright policy, authors who use figures or other material (e.g., graphics, clipart, maps) from another author or copyright holder must demonstrate or obtain permission to publish this material under the Creative Commons Attribution 4.0 International (CC BY 4.0) License used by PLOS journals. Please closely review the details of PLOS’s copyright requirements here: PLOS Licenses and Copyright. If you need to request permissions from a copyright holder, you may use PLOS's Copyright Content Permission form.

Potential Copyright Issues:

Figure 1: please (a) provide a direct link to the base layer of the map (i.e., the country or region border shape) and ensure this is also included in the figure legend; and (b) provide a link to the terms of use / license information for the base layer image or shapefile. We cannot publish proprietary or copyrighted maps (e.g. Google Maps, Mapquest) and the terms of use for your map base layer must be compatible with our CC-BY 4.0 license. 

Additional Editor Comments (if provided):

The authors are required to address the comments provided by the reviewers.

Reviewers' comments:

Reviewer's Responses to Questions

**Comments to the Author**

1. Does this manuscript meet PLOS Global Public Health’s publication criteria? Is the manuscript technically sound, and do the data support the conclusions? The manuscript must describe methodologically and ethically rigorous research with conclusions that are appropriately drawn based on the data presented.

Reviewer #1: Yes

Reviewer #2: Yes

2. Has the statistical analysis been performed appropriately and rigorously?

Reviewer #1: Yes

Reviewer #2: N/A

3. Have the authors made all data underlying the findings in their manuscript fully available (please refer to the Data Availability Statement at the start of the manuscript PDF file)?

Reviewer #1: No

Reviewer #2: Yes

4. Is the manuscript presented in an intelligible fashion and written in standard English?

Reviewer #1: Yes

Reviewer #2: Yes

5. Review Comments to the Author

Reviewer #1: Authors receive praise for creating comprehensible and well-organized pieces.

How the relevant discussions are presented offers a valuable division of sections in which information has been appropriately conveyed, enhancing the quality of this work. This is further complemented by the successful execution of the Community Health Worker (CHW) project.

Reviewer #2: Dear Authors,

I want to commend you on the extensive work you have undertaken to examine Community Health Worker (CHW) programs supported by the Canadian Red Cross (CRC) in Africa. Your research is highly relevant, especially considering the critical role that CHWs play in healthcare delivery in low-resource and humanitarian settings. The manuscript does an excellent job of outlining the methodology and the breadth of projects considered. The use of the READ approach for document analysis adds a robust framework to your study. Furthermore, your findings contribute valuable insights into the facilitators and barriers affecting CHW programs, filling an essential gap in the literature. However, there are some areas where the manuscript could be improved for clarity, depth, and impact:

Abstract

• Please consider including the number of reports that were analysed in your document analysis. This would offer readers an idea of the study’s scope and the weight of the evidence presented.

• Another area that could make a stronger impact is the specificity of your findings. The abstract offers a broad summary, but highlighting one or two key, concrete findings could draw more attention and add more value to the research.

• Regarding your study objective, the aim to explore "key findings and key lessons learned" from CRC-supported CHW programs is clear but could benefit from greater specificity. What makes a particular finding key? You can be clear on whether you looking at health outcomes, program sustainability, cost-effectiveness, or some other measure of success? Providing a more targeted focus could add rigor to your study and make the objective more transparent to the reader.

Introduction

• Line 87-92 : the authors transition from discussing the role of CHWs in humanitarian crises to specific activities in Africa. While some regions in Africa do experience such crises, the continent is diverse, and not all areas are affected in the same way. Specifying which countries or regions in Africa you are referring to could add more nuance and accuracy to your discussion.

• Line 100-103: the authors argue about "important knowledge gaps" regarding the implementation of CHW programs in humanitarian, low-resource, and conflict settings. While you mention that "important knowledge gaps remain," the introduction does not delve deeper into what these gaps specifically entail or cite existing research that has attempted to address them. Given that your study aims to contribute to filling these gaps, a more comprehensive review of current literature would significantly strengthen your introduction. For example, a study by Miller et al. (2020) reviewed the evidence base on CHWs in humanitarian settings and highlighted key challenges and facilitators that could be considered when implementing CHW programs in such contexts. Additionally, a systematic review by Werner et al. (2023) provides valuable insights into the effectiveness of CHW-delivered healthcare in post-conflict settings. Integrating the findings from these systematic reviews into your introduction could offer readers a richer context and better understanding of the specific knowledge gaps your study aims to fill. This would not only solidify the rationale for your research but also potentially strengthen its impact and relevance.

o Miller NP, Ardestani FB, Dini HS, Shafique F, Zunong N. Community health workers in humanitarian settings: Scoping review. J Glob Health. 2020 Dec;10(2):020602. doi: 10.7189/jogh.10.020602. PMID: 33312508; PMCID: PMC7719274

o Werner K, Kak M, Herbst CH, Lin TK. The role of community health worker-based care in post-conflict settings: a systematic review. Health Policy Plan. 2023 Feb 13;38(2):261-274. doi: 10.1093/heapol/czac072. PMID: 36124928; PMCID: PMC9923383.

Methods

The authors did well in providing a comprehensive outline of how the analysed the retrieved documents. The READ steps used is well articulated. The authors may consider the comments below as areas that could be improved:

- The methods section would benefit from the authors stating the number of documents that were analysed. This will give readers an idea of the scale and scope of the study.

- It is a good practice to note that each project was extracted by two team members. The authors need to include a statement on how discrepancies between them were resolved. Was it through discussion? Or a third reviewer?

- While ethics approval is mentioned, the manuscript could expand on any additional ethical considerations, especially given the sensitive nature of humanitarian settings. The authors mentioned keeping the data in a secured Team’s Channel. I think this ensured confidentiality of the data. Were there other measures?

Findings

- Line 268 - Your manuscript mentions 'Data extraction of the studies' but it seems the primary data source are reports supported by CRC. It would be more accurate to consistently refer to these as 'reports' to avoid any confusion about the type of data being analyze.

- Line 224- 226: the statement that the projects reached 'several countries across Africa' could be clarified to be more informative by specifying the exact number of countries involved in the projects. I would also like to see a clearer communication of the geographical limitations of the project’s implementation.

- Your presentation of the data in table 1 is quite clear. I noticed that the multi-country projects are grouped together in the table. Could you clarify the reason for this approach? Was it possible to include subtotals for these multi-country projects? This would offer a more consolidated view of the impact across individual countries involved in these multi-country initiatives.

- Regarding barriers and facilitators, the authors seem to treat Africa as a monolithic entity when discussing challenges like political unrest or climate conditions. Given that the study encompasses multiple countries in Africa, it would be beneficial to include a comparative analysis. Specifics on how different environmental or political conditions affected CHW programs in each country would give a nuanced view and avoid over-generalizations. Could you specify which countries faced what kinds of challenges to provide a more comprehensive understanding?

6. PLOS authors have the option to publish the peer review history of their article (what does this mean?). If published, this will include your full peer review and any attached files.

**Do you want your identity to be public for this peer review?** For information about this choice, including consent withdrawal, please see our Privacy Policy.

Reviewer #1: No

Reviewer #2: **Yes: **Isaac Amankwaa (PhD,RN)

---

## [Decision Letter · Decision Letter 1]

18 Dec 2023

Engaging Community Health Workers (CHWs) in Africa:

lessons from the Canadian Red Cross supported programs

PGPH-D-23-01553R1

Dear Ms. Idriss-Wheeler,

We are pleased to inform you that your manuscript 'Engaging Community Health Workers (CHWs) in Africa:

lessons from the Canadian Red Cross supported programs' has been provisionally accepted for publication in PLOS Global Public Health.

Best regards,

Ama Pokuaa Fenny, Ph.D

Academic Editor

Reviewer Comments (if any, and for reference):

Reviewer's Responses to Questions

**Comments to the Author**

1. If the authors have adequately addressed your comments raised in a previous round of review and you feel that this manuscript is now acceptable for publication, you may indicate that here to bypass the “Comments to the Author” section, enter your conflict of interest statement in the “Confidential to Editor” section, and submit your "Accept" recommendation.

Reviewer #1: All comments have been addressed

Reviewer #3: All comments have been addressed

2. Does this manuscript meet PLOS Global Public Health’s publication criteria? Is the manuscript technically sound, and do the data support the conclusions? The manuscript must describe methodologically and ethically rigorous research with conclusions that are appropriately drawn based on the data presented.

Reviewer #1: Yes

Reviewer #3: Yes

3. Has the statistical analysis been performed appropriately and rigorously?

Reviewer #1: N/A

Reviewer #3: N/A

4. Have the authors made all data underlying the findings in their manuscript fully available (please refer to the Data Availability Statement at the start of the manuscript PDF file)?

Reviewer #1: Yes

Reviewer #3: Yes

5. Is the manuscript presented in an intelligible fashion and written in standard English?

Reviewer #1: Yes

Reviewer #3: Yes

6. Review Comments to the Author

Reviewer #1: I appreciate you sending over the updated version. Although I also approved of the earlier draft, I think the revised manuscript is stronger. I applaud the writers' efforts, which will be useful to scholars all across the world.

Reviewer #3: The study has been reported in a way I believe it provides important information on how to address the issues emerging from shortage of health workforce manpower and the all-important universal health coverage, especially, in low-and middle- income countries. Well articulated and impressive.

7. PLOS authors have the option to publish the peer review history of their article (what does this mean?). If published, this will include your full peer review and any attached files.

**Do you want your identity to be public for this peer review?** For information about this choice, including consent withdrawal, please see our Privacy Policy.

Reviewer #1: No

Reviewer #3: **Yes: **Nnodimele Onuigbo ATULOMAH
